# Non-Healing Perianal Fistulas: A Clinical Model of Tissue Senescence Impairing Both Tissue Fibrosis and Regenerative Potential

**DOI:** 10.3390/biomedicines11020537

**Published:** 2023-02-13

**Authors:** Jason Llaneras, Caitlyn C. Belza, Samuel Eisenstein, Marek K. Dobke

**Affiliations:** 1Division of Plastic Surgery, Department of Surgery, University of California, San Diego, CA 92093, USA; 2Division of Colorectal Surgery, Department of Surgery, University of California, San Diego, CA 92093, USA

**Keywords:** tissue senescence, fibrosis, wound healing, regeneration

## Abstract

Senescent cells and fibrosis are important components that impact the regenerative capacity of skin, particularly when considering chronic non-healing wounds. Anoderm and perianal fistulas in the setting of Crohn’s disease are clinically pathophysiological extremes with consequently different healing processes which impact treatment modalities. This study describes the implications of potential senescence reversing techniques including autologous fat grafting and pharmacologic and immunomodulating agents. Given these findings, the authors propose a future direction of study involving exosomes loaded with senolytics as a method for potentially improving chronic wound healing. In conclusion, this manuscript explores the diversity of skin healing and healing outcomes which supports the future investigation of senotherapeutic agents promoting regenerative processes for non-healing wounds.

## 1. Introduction

One could argue that the cover title of *Biomedicines’* Special Issue, “Fibrosis versus Regeneration of Skin”, is unfitting. Wound healing involves both fibrotic and regenerative components, and clinically exhibits different proportions of both mechanisms. Fibrosis is at one end of the spectrum (Figure 1), while the regenerative capacity of the skin, which depends on multiple elements, is at another end of the spectrum (Figure 2A,B). Surgical or traumatic acute wounds are expected to progress through the phases of normal healing, which typically result in a soft tissue defect. Normal wound healing consists of three overlapping stages: inflammation (4–6 days), cellular proliferation (4–24 days), and remodeling (21 days–2 years). Each phase is characterized by a variation in the type of cellular involvement and differences in the profile of cytokines recruited. Chronic wounds are defined as defects to the tissue that fail to progress in healing over the expected normal time frame, which is approximately four weeks. Typically, the fibrotic and regenerative arms of healing arrest in the inflammatory phase [1]. As the preliminary phase of wound healing, this phase features coagulation via platelet aggregation and the clotting cascade followed by recruitment of inflammatory cells. The inflammatory cells are recruited by various products of platelet degranulation (i.e., tissue growth factor beta), bacterial degradation (i.e., lipopolysaccharides) and the complement cascade which produces C5a. They key inflammatory cellular players include neutrophils, macrophages, monocytes, lymphocytes, and mast cells. 

Given the intricate relationship between basic science and clinical models, these are both important factors contributing to the discussion of this volume’s main topic of skin fibrosis versus regeneration. Certainly, lessons learned from extreme cases of fibrosis and regeneration may have a significant impact on the research and treatment of standard acute and chronic skin lesions. One example of these extremes is scarless fetal wound healing. A transitionary phase between skin healing without fibrosis and healing with scar formation exists, and in humans it occurs gradually after 24 weeks of gestation [2]. This favorable pattern of healing persists for a while, which is evidenced by the unheard-of hypertrophic scarring in patients with early cleft lip repair. It is hypothesized that the absence or minimal acute inflammatory response in fetal wounds determines the trajectory of subsequent wound healing. It is postulated that this is related to differences between fetal and adult thrombocyte function, particularly within the cellular signaling pathway [3]. In addition, there are secretory differences between fetal and adult fibroblasts leading to alterations of the extracellular matrix. One might also consider the idea that fetal skin is composed of a higher ratio of type III collagen to type I collagen when compared to the adult skin. Finally, fetal skin has fewer cytokines promoting inflammation, which minimizes the development of the fibrotic response in utero [4].

In postnatal life, keratinocytes are present at superficial defects of the skin and retain the capacity to fully regenerate. However, in certain conditions (e.g., epidermolysis bullosa) there is a loss of the integrity of the skin epidermis, its ability to regenerate or differentiate, and its normal mechanism of compensation for daily wear and epidermal defect healing (Figure 2A,B). There are multiple signaling pathways regulating the ability of keratinocyte stem cells to proliferate and re-epithelialize superficial defects [5].

It has been shown that deeper dermal and subcutaneous layers are more likely to heal with fibrosis and scar formation (Figure 3). Insufficient vascularity and a senescent microenvironment are two important denominators in failed healing [6].

Regenerative repair involves the replacement of tissues with components comparable to the original tissues. In regeneration the aim is to restore or reestablish a normal, harmonious whole—an elegant interweaving of tissue elements. However, this is not always the case. We develop a scar when the fibrotic components prevail, and the regeneration of fibroblast exceeds the reformation of other native tissue components (e.g., blood vessels) [7] (Figure 4). We would therefore argue that the cover title should be “From Regeneration to Fibrosis of Skin”.

Biologically, fibrosis is part of skin regeneration. Given the dermatologist or plastic surgeon perspective, fibrosis is usually linked with an untoward outcome of healing [8]. Conversely, scarring of the dermis following superficial wounds can result in skin tightening and may be considered a cosmetic benefit [9]. In skin wound healing, the matricellular protein CCN1 can induce fibroblast senescence and thereby reduce fibrosis via activation of the DNA damage response and reactive oxygen species signaling [10]. However, the formation of scar tissue is classically perceived as detrimental for the overall outcome from both a cosmetic and functional standpoint. In some situations, the morbidity associated with a wound is so profound that healing of the defect by either fibrosis, epithelial regeneration, or both is the best clinical course of action and often appreciated by the patient [11].

Senescent cells and fibrosis are important components of the healing and aging processes. One of the hallmarks of chronic non-healing wounds is cellular senescence. Senescent cells are alive and metabolically active but non-proliferative. The pathogenesis of chronic skin wounds is complex and may involve many different molecular and cellular pathways. Exceedingly proliferative fibroblasts may create an imbalance between the healing trajectory of different organ (e.g., skin) components by forming scar on the expense of regenerative processes.

Why was the model of two clinically relative pathophysiological extremes, anoderm and Crohn’s disease, selected for review to address the issue of skin fibrosis versus regeneration? These were chosen to increase evidentiary weight, given that wound healing research models are typically limited and focused either on fibrosis or regeneration. Measuring the immediate clinical utility of diverse basic science research observations is difficult. Across most realms of scientific discovery, researchers are posed with the challenge of translating benchwork to bedside use. Additionally, clinical practice and management strategies typically require a combination of problems that must be addressed. One such example, which demonstrates a multiplicity of healing problems, is anodermal lesions during the course of Crohn’s disease.

That being said, there have been major recent advancements in innovative research, which have changed the game for clinical management of wound healing. Some note worth examples include transforming growth factor beta (a central factor of fibrosis), controlling collagen production, and understanding myofibroblast differentiation leading to matrix accumulation and contraction. Research has also advanced the conceptualization of metastatic spread of cancer cells with pleiotropic signaling which contributes to oncological drug resistance. Perhaps all wounds ranging from fetal skin wounds, to acute skin wounds in adults, to chronic lesions, exhibit a different degree of inflammation, proliferation, regeneration and fibrosis. However, the chronic wound model allows for identification of factors leading to healing pathology. This consequently lends to the development of novel and potentially more precise treatments through a targeted approach to drug discovery.

Perianal Crohn’s Disease (PCD) is a major cause of morbidity in patients with Crohn’s, affecting up to 30% of patients with luminal disease. This disease, which may includes skin tags, fissures, abscesses, fistulae, and stenosis often leads to significant pain, discharge, bleeding, itching, and ultimately incontinence. Patients with PCD have greater challenges in obtaining clinical remission than those with purely luminal disease. In particular, perianal fistulizing disease has proven to be one of the greatest treatment challenges in maximizing patients’ quality of life. Such wounds require simultaneously effective fibrosis and regenerative mechanisms to heal. These fistulas offer a model of impairment involving both fibrotic and regenerative components. In addition, clinicians are aware that this healing impairment in Crohn’s disease patients is at the extreme level [12]. Healing impairment in Crohn’s disease is not only extreme but also truly multifactorial. Therefore, the review of healing of anodermal lesions under this disease pathophysiology appeared to be a valid and interesting pathology model for the review. The knowledge to be gained from this example is important for surgeons, but also pertains to researchers across many domains of study.

In healthy individuals, the anoderm is known to heal rapidly despite a seemingly unfavorable healing environment which is composed of constantly moving tissue layers and heavy loads of microbial organisms. On the other hand, perianal fistulas in the setting of Crohn’s disease are particularly prone to non-healing due to the nature of the disease itself [13,14]. Histological findings in non-healing anal fistulas include major defects in the stratified squamous epithelium. The absence of signs of epithelialization would suggest that the predominant healing impairment involves regenerative components [15].

To date biologic therapy has been the medical mainstay of treatment in this patient population. When considering the options, the most effective treatment involves antibodies directed against TNF-a, however, management can also include anti-integrin therapy, as well as medicines targeting various interleukins and the JAK signal transduction cascade [13]. “Excisional/tissue rearrangement” surgery supposed to convert wound from chronic to acute. However, even in the best of circumstances, employing these medications with effective curettage and drainage, clinical healing rates are generally below 50% with early recurrence rates also approaching 50% in the group which achieved clinical remission [12,13]. Management of the local sepsis with the previously mentioned curettage and seton drain placement is considered necessary to optimize the local microenvironment for fistula resolution.

There are also numerous surgical options for patients with Crohn’s fistulae, primarily because none have proven to be particularly effective or superior to the others. That being said, there is certainly evidence that surgery, in the setting of medical management, increases the likelihood that patients will obtain clinical closure of their fistula tracts [12]. The primary tenets of most surgical options include closure of the internal orifice, ligation of the fistula tract, and preservation of continence via minimal transection of the anal sphincter musculature. Because of the inflammatory nature of the disease only a minority of patients will even be considered candidates for some of the more complex repairs given that the surgical wounds are at risk of ineffective healing should they be created in the setting of poorly controlled inflammatory disease. This, in turn, places the patient at a greater risk for a variety of complications and thus care needs to be taken when considering which patients may ultimately be a surgical candidate. The potential for local sepsis must also be controlled via effective drainage prior to considering repair. Usually, the combination of seton drainage and medical management means that patients must often wait great periods of time between the initial development of their fistula and the attempts at definitive surgical management. The expectation is that with time, control of sepsis, and inhibition of the inflammatory process, that the local fistula microenvironment will more closely represent the M2 phenotype which would potentially lead to higher fistula healing rates.

Clearly the essential processes of “fibrosis” to obliterate the fistula tract and “regeneration” to restore the defect by epithelialization are uniquely intertwined and tissue senescence appears to be the key factor yielding non-healing tissue. In addition, the anoderm skin resides in the area between the dentate line and the area around the anal verge which is composed of non-keratinized squamous epithelium. This region has a relatively low stem cell content, as opposed to skin elsewhere which contains hair follicles and glands [16]. The relative absence of structures such as keratinocytes, which source multipotent stem cells, strengthens the notion that anoderm healing should be a model for investigation of fibrosis vs. regeneration [17]. Therefore, studies evaluating the efficacy of different treatment modalities for lesions located in these extreme conditions (i.e., in the anus, and in Crohn’s disease) may be needed to gain a better understanding of skin regeneration, bioengineering of skin substitutes, treatment of skin diseases and cosmetology [18].

Most vascular changes in Crohn’s disease are either degenerative or inflammatory in nature resulting in reduced tissue perfusion [6,19] (Figure 5). Surgical intervention (e.g., fistulotomy) removes locally diseased tissue but does not reverse the underlying disease process. Local administration of anti-inflammatory agents (e.g., TNF-alpha antibodies) reduces signs of inflammation but rarely results in permanent fistula healing. The use of senolytic agents (e.g., nutritional supplement Quercetine) or autologous fat transfer with stem cell fraction “controlled” by senescent cells appears promising as they target multiple senescence-associated molecular pathways [13,20].

## 2. Senescence Reversing Modalities

### 2.1. Autologous Fat Grafting

In recent years, topical transplantation of autologous fat tissue for the treatment of complex wounds, tissue rejuvenation and contouring has become increasingly popular [21]. Several studies have shown promising outcomes when lipoaspirate is used to treat chronic wounds such as diabetic ulcers, radiation lesions, and non-healing burns. Even though the clinical effects of fat transfers are encouraging, the underlying pathway of this senolytic effect is still a matter of investigation. Current theories suggest that this mechanism is multifactorial. More dramatic results are associated when the stromal vascular fraction (stem cell to adipocyte ratio) is increased. Some studies have shown that supplementing the injected fat and the use of different fat processing techniques have been able to increase this ratio [22]. Extrapolating on this, our research team found surprising results when investigating how adipocyte derived stem cells are influenced by chemotherapy drugs in breast cancer patients. We observed moderate changes regarding cell viability, cell count, cell populations and receptor profile by flow cytometry, and colony forming units with rebound after systemic chemotherapy.

This should warrant similar investigations in Crohn’s disease patients [23]. One of the plausible explanations of the diverse results of fat grafting may be related to changes in macrophage polarization and a shift toward an anti-inflammatory phenotype leading to reduction in fibrosis and improved fat graft retention [24]. Knowledge of the metabolic shift from a catabolic state to an anabolic state (from senescence to proliferative profile) could impact the sequencing and timing of pharmacological treatment. Some propose that change of the local hormonal milieu though the delivery of leptins may play a significant role, especially from an angiogenic standpoint [25]. There is also data supporting the hypothesis that the local cytokine milieu serves as critical factor in the efficacy of local stem cells treatment [16,21]. The multipotent capacity of fat cells may be an instrumental aspect of enhanced healing through multiple mechanisms. For instance, the capability of adipocytes to differentiate into dermal fibroblasts may accelerate formation of “granulation tissue” and stimulate the fibrotic component of healing. Additionally, keratinocyte-like cells have been found to maintain the capacity to form epidermis, which may exemplify a regenerative-type of healing [26].

### 2.2. Clinical Perspective

Clinical outcomes support expectations derived from experimental investigations and observations of chronic wounds other than anodermal wounds. While our thirty-four Crohn’s disease patients did experience decreased success rates when compared with non-Crohn’s disease patients, they still had 79% improvement and 49% closure, with 35% recurrence, superior to the recurrence rate of 44% for infliximab and non-cutting seton placement when autologous fat grafting was performed [27]. This is also comparable to reported closure rates of around 58% in endorectal advancement flap procedures and 48% for ligation of intersphincteric fistula tract procedures in Crohn’s disease [28].

### 2.3. Pharmacological and Immunomodulating Agents

Pharmacological agents downregulate host pro-inflammatory mediators (e.g., tumor necrosis factor alpha) and upregulate anti-inflammatory or immune-regulatory factors both of which improve the healing potential of anoderm fistulas. There has been reported differences in gene expressions between patients who have had a clinical improvement. Certain genes that favor a positive clinical response include TNFAIP6 and IL-11. Extrapolating this data to general skin biology suggests that the balance between regenerative and fibrotic potential is genomically determined and that the development of personalized protocols for skin therapeutics will certainly require a personalized approach [8,13,29]. Additionally, inflammatory bowel disease lesions may not heal (e.g., do not re-epithelialize) because of epithelial cell telomere dysfunction. Observations examining experimental senescence models with telomere dysfunction point to the therapeutic potential of pharmacological interventions with telomerase reactivation and suppression of DNA damage signaling which restores epidermal regenerative capacity [19,30,31]. Persistence and frequent recurrence of fistulas supports the notion that the disease is more related to inflammatory rather than infectious factors. Observed reductions in the macrophage M1/M2 ratios in anal fistula patients supports this notion. The M2 anti-inflammatory phenotype supports neovascularization and fat graft retention. Reducing this ratio would consequently support both fat and stem cells grafting to tissue affected by fistulization. The administration of agents that could change the cellular signaling pathway of macrophage polarization, would ultimately result in wound healing [24,32].

In general, lessons learned from research on tissue senescence have an application beyond fibrosis versus regeneration in “extreme” conditions such as anal fistulas, diabetic foot ulcers and epidermolysis bullosa. Pivotal aspects of the pathophysiology of these conditions allows for easier identification and understanding of therapeutic opportunities. The creation of inflammation and fibrosis, which reduce immune microenvironments, and support the regenerative capacity of skin components is shared by pathological and “non-pathological” conditions (i.e., skin aging) with concepts of comprehensive, multiprong, medical and cosmeceutical interventions, respectively [33].

## 3. Future Directions

One should not speak about the future without knowledge of the past. Many traditional, sometimes considered old fashioned or under-investigated natural remedies enjoy somewhat of a renaissance of applications when new evidence is brought to light [34]. For example, *Aloe vera* has recently been the subject of many clinical trials with a goal of assessing its potential in “official medicine”. Evidence suggests that aloe is effective in restoring skin integrity and promoting the healing of chronic wounds [35]. The next step in such research is extraction and purification of active *Aloe vera* factors and investigation of their mechanisms action which may impact skin fibrosis and regeneration. The results of this line of study may provide expectedly or unexpectedly, a novel and clinically useful tool. One of the best examples of the transition from ancient application based on empiric observations to evidence based *Pharmacopea* involves the history of honey for wound management. Today, findings suggest that honey may be applied in a clinical setting for its ability to reverse tissue senescence and promote healing of stubborn, non-healing wounds [36].

Some of the advanced wound care modalities that have been developed or are undergoing the development process include bioengineered allogeneic cellular therapies, refining of stem cell based interventions, xenograft cellular matrices and growth factors.

Extracellular vesicles (exosomes) which may contain cellular contents including proteins, lipids and nucleic acids play a role in inter-cellular communication and may release senescence secretory phenotype factors. Conversely, exosomes may be loaded with senolytics (e.g., Serpins, agents regulating macrophage polarity) and promote healing (Figure 6). Increasing evidence suggests that extracellular vesicles (EVs) mediate indirect signaling between cell types in wound healing [37]. Studies of pathways regulating EV biogenesis have demonstrated that the cellular source of EVs affects their formation, payload, and biological activity in the wound bed.

For example, platelet-derived EVs promote coagulation in hemostasis, while neutrophil-derived EVs regulate the expression of adhesion factors on the endothelium. As a wound transitions from the inflammatory phase to the proliferation and re-epithelialization phases, macrophage-derived EVs drive macrophage polarization to an anti-inflammatory phenotype (and possibly senescence-associated secretory phenotype) and mediate crosstalk signaling with wound-edge fibroblasts and keratinocytes to promote wound closure. The resolution of a wound after the inflammatory phase (wound closure) involves re-epithelialization through keratinocyte migration and remodeling of granulation tissue into a more permanent extracellular matrix through expression of collagen, proteoglycans, and regulation of the secretion of proteases. Exogenously derived senolytics-loaded EVs applied onto wounds or into wound surroundings could reverse senescence and trigger wound closure. Notably, exosomes from adipose stem cells promote chronic (e.g., diabetic) wound healing [38].

Another interesting, somewhat futuristic, approach involves an intraoperative alteration of the skin wound milieu. To reduce inflammation and stimulate the commencement of an “almost immediate” regenerative process of skin healing while reducing fibrosis, some have employed an intraoperative, intradermal injection of stromal cell fraction (derived from autologous fat tissue). These preliminary clinical experiments need more follow up and experimental reaffirmation before conclusions regarding efficacy of such s concepts can be drawn [39]. While there are many examples of faulty regeneration of skin components (Figure 2), clinical practice also demonstrates pathology involving excessive skin regeneration and the need for its reversal. Regarding wound healing, hyperproliferation and fibrotic conditions exist separate from oncologic conditions where stem cell-associated pathways lead to aberrant cell self-renewal and differentiation [40]. For cases of excessive fibrosis, there is an interesting line of study involving autologous fat grafting, which appears to be beneficial for reversing skin fibrosis. The mechanism by which fat grafting contributes to the clinically observable improvement in healing (in cases when fibrosis is clinically unwelcomed) has not been clarified; however, there is much anticipation for further research examining this potentially useful application [26,41].

## 4. Other Implications on Wound Healing

Space obliteration is expected and clinically observable when fat is used as a volumizer (and spacer to bring the fistula walls into direct contact by external pressure) regardless of the impact of stromal. It is plausible that this adjunct mechanism leading to fistula healing is similar to the “internal obliteration” exerted by fibrin glue administration into an abraded fistula lumen. An intriguing question arises about whether tissue scaffolding capacity and injectable volume retention in a low (or absent) stem cell environment is similar to other areas of the body where injectable fillers are used for cosmetic purpose? Additionally, one could ponder if these fillers can be used to enhance wound healing in this setting [42].

The effects of skin microneedling in the cosmetic surgery arena is an example of “beneficial” fibrosis. Fistula “micro- or macroneedling” refers to intentional surface injury with mechanical creation of raw surfaces. This precipitates fibrosis and potentially a scar which helps to obliterate the fistula lumen. Stimulation of inflammation and healing de novo may also help to reverse chronic “burnt out” senescent Crohn’s disease [14].

As stated in the introduction, conditions that exhibit extreme fibrosis and regenerative reactions, or their absence of these mechanisms are valuable learning tools for researchers and clinicians. Understanding diverse and intricate wound etiologies provides a useful experience that may be applied to the management of both common and rare conditions. In the embryonic stage of life, factors which regulate cell growth, motility, and differentiation by activating diverse signaling cascades (Wnt: wingless-related integration site and pathways) seem to determine overall health in postnatal life. These factors may be associated with the development of specific disease and healing patterns [43]. Similarly, JAK-STAT signaling, and other intercellular communication mechanisms are involved in immunity, cellular division, and apoptosis which are critical aspects of chronic wound healing during adult life [44]. Pyogenic sterile arthritis, pyoderma gangrenosum, and acne (PAPA) and autoinflammatory variants including pyoderma gangrenosum, acne and suppurative hidradenitis (PASH) are notorious for impaired healing and provide a different perspective of skin fibrosis versus regeneration than the examples illustrated in this review. These additional conditions share common pathologic processes including genetic mutations (i.e., gene PSTPIP1 affecting pyrin), errors in tyrosine phosphatase regulatory processes, and over-activation of the innate immune system leading to increased production of the interleukin-1, tumor necrosis factor alpha and neutrophilic skin inflammation [45].

## 5. Conclusions

The general awareness of the pathophysiologic processes affecting skin healing by promoting regeneration and inhibiting fibrosis is growing. This review of anoderm wound biology supports the notion that it can serve as a model of nearly fibrosis-free skin healing in a stem cell deficient environment. Interestingly, this model does not replicate healing features seen in other stem cell deficient defects (nail dystrophy models requiring destruction of the proximal matrix where stem cells reside) resulting in dominant fibrosis [46]. However, research of processes in this environment with extreme intrinsic and extrinsic factors may help to understand the diversity of skin healing and healing outcomes. Consequently, it should not come as a surprise that senotherapeutic medications (systemic and topicals), as well as cosmeceuticals, are being thoroughly investigated for their use in different methods of skin rejuvenation [47,48].

## Figures and Tables

**Figure 1 biomedicines-11-00537-f001:**
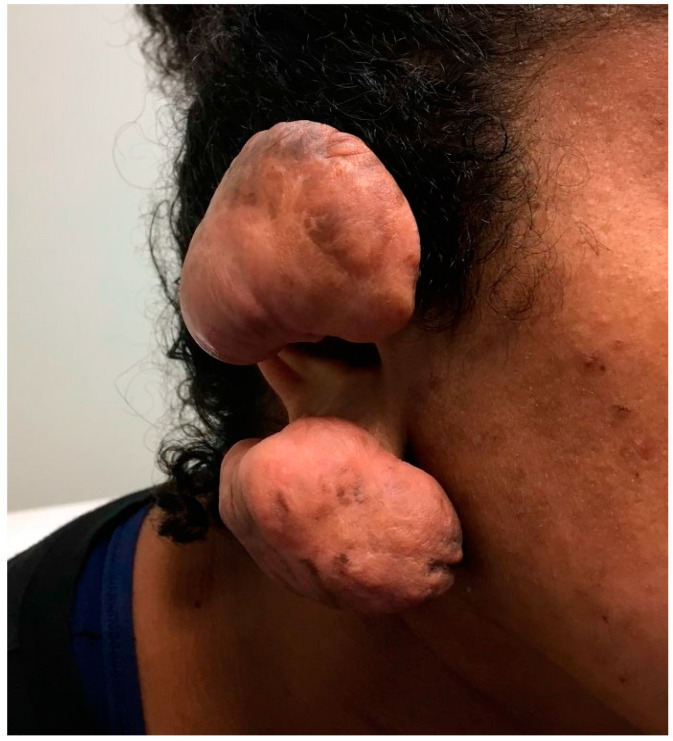
A twenty-two-year-old female with ear wounds secondary to a piercing. The patient’s ear healed with significant fibrosis and formation keloids.

**Figure 2 biomedicines-11-00537-f002:**
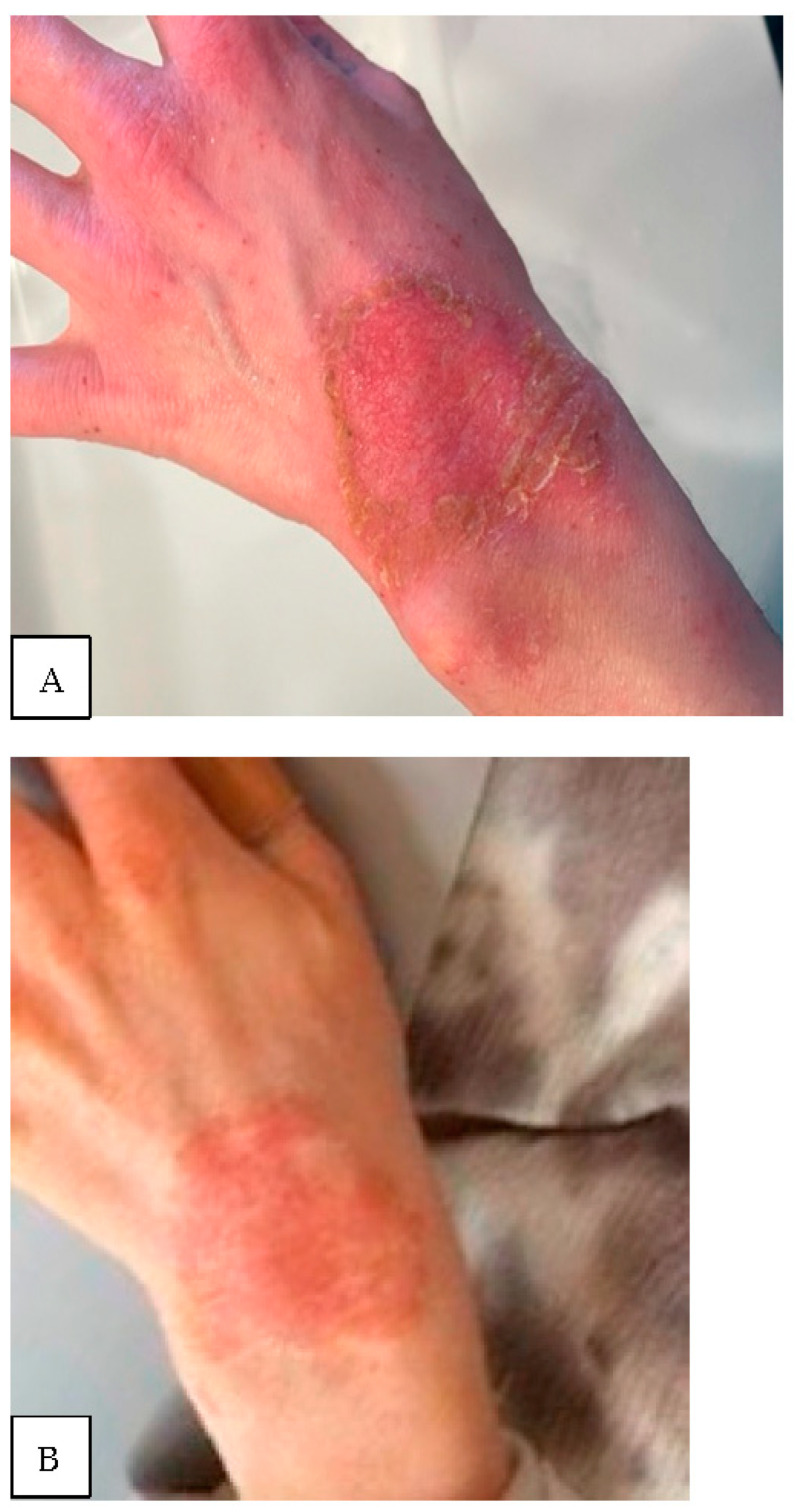
(**A**) Thirty-two-year-old female with atopic dermatitis with a two-month history of progressive, non-healing superficial wrist wound; (**B**) the same patient three months later with an extremely apparent slow rate of re-epithelialization.

**Figure 3 biomedicines-11-00537-f003:**
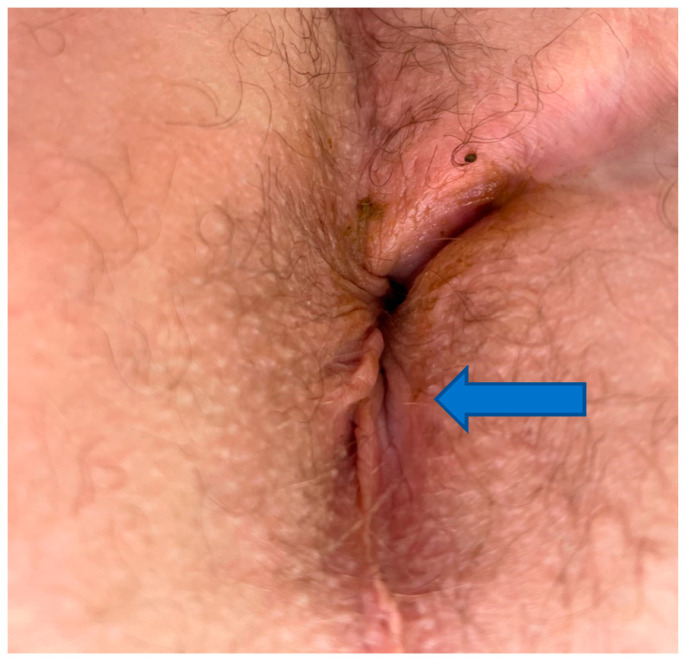
Thirty-six-year-old male with four-year history of non-healing trans-sphincteric anal fistula involving epidermis (arrow), deeper wound involving dermis (closer to the midline). Healing of such wound requires both effective fibrosis and regenerative mechanisms.

**Figure 4 biomedicines-11-00537-f004:**
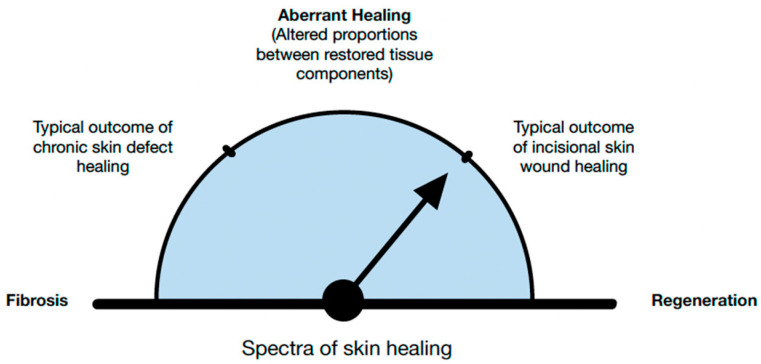
Regenerative healing is characterized by the restoration of structure, function, and physiology of damaged or absent tissue. Even under the most favorable conditions, post-fetal healing is rarely 100% regenerative.

**Figure 5 biomedicines-11-00537-f005:**
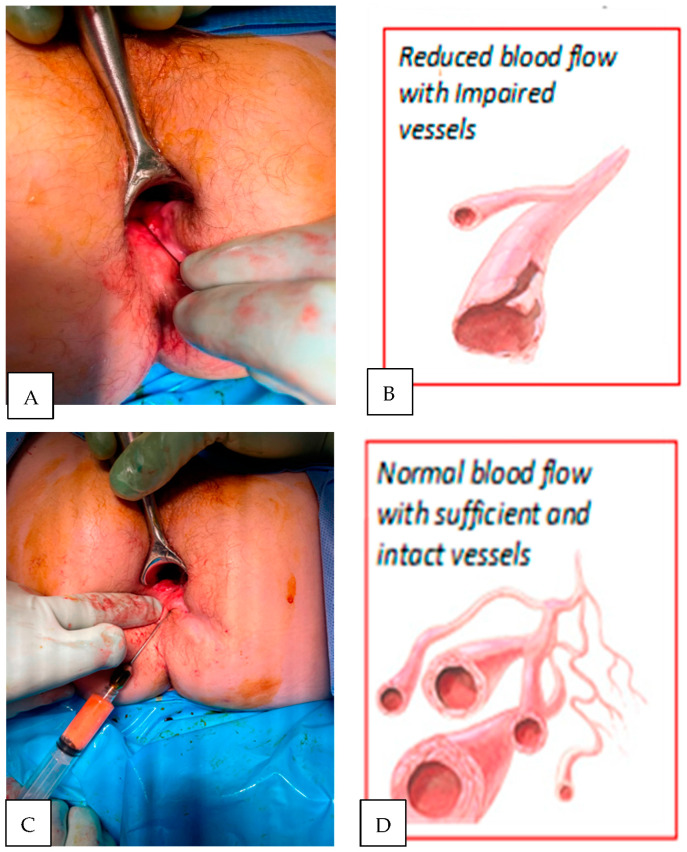
Crucial constituents for the reversal of tissue senescence and the shift of the healing processes towards regenerative the mode is initiated by peri-fistula administration of autologous fat with stromal cell fractions. Fat tissue provides a scaffold to support regenerative repair, angiogenesis, cytokines supporting cells proliferation, differentiation and helps to create environment capable of responding to anti-inflammatory drugs and biomechanical/surgical stimuli. (**A**) Non-healing fistula lesion (Crohn’s disease patient); (**B**) biopsies reveal poor microvascular network in fistula walls; (**C**) topical administration of processed (washed and filtered utilizing PureGraft System, Bimini, Solana Beach, CA, USA); (**D**) a few months later, biopsy typically reveals enriched blood vessels network.

**Figure 6 biomedicines-11-00537-f006:**
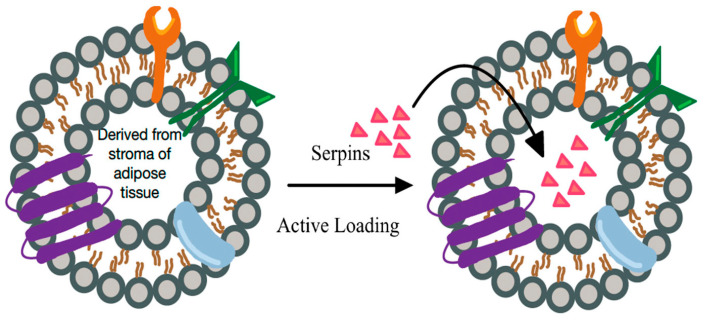
“Natural” and “engineered” extracellular vesicle (EV) based signaling. EVs loaded with specific factors for reversing senescence and stimulating healing (e.g., serpins) may promote healing in all wound healing phases: hemostasis, inflammation, proliferation, and remodeling. They may also amplify the effectiveness of other therapies targeting senescent cells.

## Data Availability

Not applicable.

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
