# Peer review of "Non-Healing Perianal Fistulas: A Clinical Model of Tissue Senescence Impairing Both Tissue Fibrosis and Regenerative Potential"

_biomedicines, 2023, doi:10.3390/biomedicines11020537_

Round 1
Reviewer 1 Report (Previous Reviewer 1)
Dear author,
the cahnges are marginal.
As surgeon, I can't see any new aspects for clincal use.
Author Response
Thank you for taking the time to review our paper. While we respect your opinion that there are no new aspects for clinical use as a surgeon, three of the four authors of this paper are also surgeons and feel otherwise. We believe that understanding the theoretical and practical problems of skin fibrosis and ability to regenerate based on very specific model is fundamental to the practice of surgery. We are grateful for your feedback and have highlighted our perspective with the addition of several paragraphs focusing on the clinician perspective that wound healing problems associated with skin fibrosis and the science of skin regeneration are highly relevant.
Reviewer 2 Report (Previous Reviewer 2)
Unfortunately, the authors did not consider my previous advice, which is to dedicate more space to wound healing, especially the differences between acute wounds and chronic wounds. Surprisingly, the authors take this issue almost for granted. In fact, reviewing the work they dedicate the space of only 3 lines and among other aspects they do not dedicate even a citation. I therefore recommend that the authors follow my suggestions and cite current refeernces (years: 2021-2022).
Author Response
As per the Reviewer’s suggestion the full text of the manuscript has been carefully edited by a native-English speaking colleague. Additionally, the manuscript was expanded to include a discussion of acute versus chronic wounds and the various stages of wound healing to address your advice. Additionally, content has been added to review more intricate aspects of healing a brief commentary on fetal healing and its association with skin fibrosis versus regeneration. We have also added multiple citations for relevant literature which has been published in recent years. We appreciate the feedback and feel that these revisions have strengthened the paper. Thank you.
Round 2
Reviewer 1 Report (Previous Reviewer 1)
Dear author,
the paper is improved in a good way.
Did you checked any syndroms in these case like PAPA, PASH are other syndroms. Often the perianal fistula is combined with Acne inversa, Colitis, other disease from rheumatic diseases. All these are related to TNF alpha and JAK pathways.
Please, can you discuss this issue in relation to the rare diseases.
You can contact me also directly to discuss the details.
After this small revision the paper could accept for publication.
Author Response
Authors would like to thank to Reviewer 1 for giving us the opportunity to add another dimension to the review article covering “Skin Fibrosis versus Regeneration”. The Reviewer’s specific suggestions have been addressed through the addition of a concluding paragraph. Although the core of our draft remains unchanged with a focus on one model of fibrosis and regeneration impairment, the addition of a paragraph about the pathophysiology of various other families of wounds and healing (genetics, signaling, auto-inflammatory conditions such as PAPA, PASH) enriches the review. We thank the reviewer for their suggestion and agree that the added perspective on tissue fibrosis and regeneration enhances the review message.
Reviewer 2 Report (Previous Reviewer 2)
The authors have answered at my questions.
Author Response
Thank you for your comment. The article again has been reviewed by native English speaking co-authors to improve the readability of the review.
This manuscript is a resubmission of an earlier submission. The following is a list of the peer review reports and author responses from that submission.
Round 1
Reviewer 1 Report
Dear Author,
I don't understand the "Clinical Model" in your paper. As a surgeon I'm looking for new models to change my strategy to improve the outcome of my patients.
You are starting with atopic dermatitis and leads to an inflammatoriy bowel disease. Anytime we find correlations between both etiologies, but in your review a can't see this relations.
Morbus Crohn is an inflammtory disease with a high evidence in different diagnostic and therapeutical options.
I can't see any new informations or game changer in your review and in the conclusion.
Reviewer 2 Report
This is a review very interesting. This manuscript explores the diversity of skin healing and healing outcomes which supports the future investigation of senotherapeutic agents promoting regenerative processes for non-healing wounds.
The only flaw I found reading this review is that the authors assume that potential readers are experts on wound healing a process not yet fully understood. It is therefore my opinion that the authors describe and then dedicate at least one paragraph to this topic, deepening the problems related to chronic wounds.